# The Development of Rucaparib/Rubraca®: A Story of the Synergy Between Science and Serendipity

**DOI:** 10.3390/cancers12030564

**Published:** 2020-02-29

**Authors:** Nicola J Curtin

**Affiliations:** Professor of Experimental Cancer Therapeutics, Translational and Clinical Research Institute, Newcastle University Centre for Cancer, Faculty of Medical Sciences, Medical School, Paul O’Gorman Building, Newcastle University, Newcastle upon Tyne NE2 4HH, UK; nicola.curtin@newcastle.ac.uk or nicola.curtin@ncl.ac.uk; Tel.: +44-0-191-208-4415

**Keywords:** PARP, drug development, synthetic lethality, clinical trials

## Abstract

The poly(ADP-ribose) polymerase (PARP) inhibitor, Rubraca®, was given its first accelerated approval for BRCA-mutated ovarian cancer by the FDA at the end of 2016, and further approval by the FDA, EMA and NICE followed. Scientists at Newcastle University initiated the early stages, and several collaborations with scientists in academia and the pharmaceutical industry enabled its final development to the approval stage. Although originally considered as a chemo- or radiosensitiser, its current application is as a single agent exploiting tumour-specific defects in DNA repair. As well as involving intellectual and physical effort, there have been a series of fortuitous occurrences and coincidences of timing that ensured its success. This review describes the history of the relationship between science and serendipity that brought us to the current position.

## 1. Introduction: Rationale for the Development of PARP Inhibitors

Poly(ADP-ribose) polymerase (PARP) inhibitors (PARPi) have been the most significant addition to the armoury for the treatment of ovarian cancer since the introduction of platinum therapy in the 1970s and represent a paradigm shift in the way cancers may be treated. This is the story of the development of the first PARPi to enter anticancer clinical trials, Rubraca®, formerly known as rucaparib or AG014699, the nomenclature used here largely reflects the name of the drug at the time of use. The development of Rubraca was the result of a substantial intellectual and physical effort by many individuals working as a team but it is hard to avoid the conclusion that luck has played a significant role, and this story describes how serendipity has contributed to the final success. This review describes the development of rucaparib in the context of other advances in the understanding of DNA repair and highlights the lucky breaks along the way.

The development of PARPi that ultimately led to rucaparib began in Newcastle University in 1990 but the story begins before that. The first observation that suggested the existence of PARP was the profound NAD^+^ depletion following exposure of cells to ethyleneimine in 1956 [1]. Coming so soon after the discovery of DNA, this effect was not attributed to the effect of DNA methylation, rather an impact on cellular metabolism. The product ADP-ribose polymers, or poly(ADP-ribose) (PAR), was identified in 1963 and at first thought to be some new kind of nucleic acid [2], with elegant experiments a few years later showing the time course of the disappearance of NAD^+^ and the corresponding appearance of polymers and nicotinamide [3]. The first suggestion of the involvement of PARP in DNA repair was made by Edward Miller in 1975 [4]. To test this hypothesis, Purnell and Whish [5] developed the first inhibitors, which were based on the catalytic mechanism of PARP and the observation that the by-product, nicotinamide, exerts some feedback inhibition. These early benzamide inhibitors included 3-amino benzamide (3AB), which is still used as a PARPi today. The pivotal study, conducted by Barbara Durkacz in Sydney Shall’s lab, was published in 1980, showing that 3AB prevented the repair of DNA and increased cytotoxicity following exposure to the DNA methylating agent, DMS [6].

## 2. Early Studies in Newcastle

Barbara Durkacz moved to Newcastle University in the mid-1980s to establish PARP-related studies, where she was joined by Mike Purnell for the synthesis of inhibitors. However, it was the appointment of Hilary Calvert as the new director of the Cancer Research Unit in Newcastle, and his collaboration with Bernard Golding, Professor of Organic Chemistry at Newcastle University, that really got inhibitor development going. They established a drug development programme in October 1990, with funding from the North of England Cancer Research Campaign. Other members of the team were Roger Griffin (chemist), Herbie Newell (pharmacologist) and me (biologist/biochemist). PARP was one of three initial targets for drug discovery, tapping into Barbara Durkacz’s expertise. A succession of post-graduate chemistry students embarked on inhibitor synthesis guided by Roger Griffin and Bernard Golding. However, to establish any kind of structure–activity relationship (SAR) a robust, reproducible and quantitative activity assay was needed and herein lay the problem. The results using a published assay [7,8] were variable in the extreme, with replicates wildly different from each other, and no better than random. Months were spent trying to generate inhibition data until finally, in exasperation, I decided to try to get to the bottom of the problem. The assay involved incubating permeabilised cells with ^32^P-NAD^+^ and broken DNA to activate the cellular PARP then precipitating the polymer with TCA, collecting it onto filters, washing surplus NAD^+^ away and counting the acid-precipitated radioactivity. Investigating each step of the assay to identify the source of the irreproducibility by taking replicates at different stages in the assay revealed that it was nothing to do with the cell permeabilisation or collection of the product but was the reaction itself. A eureka moment led me to discover a very simple solution: switching to plastic tubes. Because of the charged nature of NAD^+^ and the polymer we had been siliconising glass tubes for the reaction to negate the charge on the glass and prevent binding. It turned out that the siliconising agent itself was causing highly variable interference with the activity of the enzyme. By switching to plastic this interference was removed but another problem was identified, i.e., the poor thermal conductivity of the plastic, which was overcome by a pre-warming step. Thereafter, this assay provided the robust, reproducible and quantitative data needed for SAR generation. This is the first example of the synergy of science and serendipity (#1). 

The next example of serendipity (#2) came on the chemistry side. Inhibitors were designed to incorporate the nicotinamide pharmacophore, which was thought to make major interactions within the catalytic site of PARP. Studies with other NAD-dependent enzymes suggested that binding was favoured when the carboxamide was anti to the aromatic ring. In 3AB, it can freely rotate to the cis-orientation, meaning that binding is relatively weak (Ki = 10 µM, IC50 = 30 µM). Parallel SAR studies led by Judy Sebolt-Leopold at Warner Lambert and an “analogue by catalogue” approach in Kunihiro Ueda’s lab identified that compounds where the carboxamide group was held anti to the aromatic ring, by constraining it through incorporation into a ring structure, had increased potency [9,10]. Our approach was to hold the carboxamide in the anti- conformation through hydrogen bonding with an oxazole or imidazole group. However, during the synthesis of the target compound, 2-methylbenzoxazole-4-carboxamide, a molecular rearrangement occurred generating 8-Hydroxy-2-methylquinazolin-4[3H]-one, NU1025, where the carboxamide was constrained in the desired orientation through incorporation into a ring [11] (Figure 1). NU1025 (Ki = 48 nM IC50 = 400 nM) had comparable activity to compounds identified by the Sebolt-Leopold and Ueda groups and was substantially more potent than the compound we had been aiming to make, which had an IC50 of 10 µM. This lucky accident therefore gave us our first hit compound and allowed us to explore the cellular effects of PARP inhibition.

My first PhD student, Karen Bowman, was tasked not only with determining the potency of the compounds against PARP activity (aided and abetted by Louise Pemberton who also synthesised some of the early inhibitors) but also with evaluating the ability of NU1025 and the benzimidazole, NU1064 (2-methylbenzimidazole-4-carboxamide, IC50 = 1.1 µM) to increase the cytotoxicity of a variety of anticancer drugs. She confirmed that PARP inhibition increases the cytotoxicity of DNA alkylating agents using temozolomide and its active derivative, MTIC. She identified that it was the DNA repair phase that was critical, rather than the DNA damage induction phase as NU1025 was as active when it was added after a 20 min pulse with MTIC as when the 2 compounds were given simultaneously. Karen also confirmed the radiosensitisation data that had been generated with the early benzamides. She showed that PARP inhibition was more effective at preventing recovery from potentially lethal damage than radiosensitising exponentially growing cells. Potentially lethal recovery is when cells are given a high dose of irradiation and their survival is estimated immediately or after a recovery period where they are allowed to repair the DNA damage. This is significant because the non-replicating fraction of a tumour is generally radioresistant. We were unable to see any sensitisation effect of NU1025 or NU1064 on antimetabolite-induced cytotoxicity [12]. Using NU1025, we were the first to demonstrate that PARP inhibition increased the DNA damage and cytotoxicity caused by topoisomerase I poisons, but not topoisomerase II poisons [13]. We continued synthesising new compounds and testing them for PARP activity with the next major step in potency being the synthesis of the benzimidazole NU1085 (2-(4-hydroxyphenyl)-benzimidazole-4-carboxamide Ki = 6 nM, IC50 = 80 nM) by Sheila Srinivasan [14]. Since our compounds were numbered sequentially, this means that our 25th compound NU1025 was 200x more potent than 3AB and our 85^th^ compound was 1000x more potent.

The generation and characterization of PARP-knockout mice that both were viable and fertile by three groups independently in the late 1990s [15,16,17], initially caused us to think that PARP was non-essential, and so inhibiting it would not compromise the viability of cancer cells. However, I recalled the data of Satoh and Lindahl [18] demonstrating that nuclear extracts depleted of PARP were able to repair nicked DNA but nuclear extracts with PARP but incubated either in the absence of NAD^+^ or presence of 3AB could not. These authors proposed that binding of PARP to the DNA break caused an obstruction unless it could auto poly (ADP-ribosylate) and dissociate to allow access of repair proteins to fill in and re-seal the gap. This was the first demonstration of PARP “trapping”, although this phrase was not coined until at least 20 years later [19]. These observations meant inhibited PARP was likely to be more effective in inhibiting DNA repair than no PARP enzyme. The ability of cells derived from these mice to generate PAR led to the discovery of PARP2 [20] and subsequently other PARPs by structural analogy [21], with the original most abundant PARP subsequently being called PARP1. It is important to note here that PARP1 is the most abundant PARP and most important in terms of DNA repair, with PARP2 having a similar role. Most inhibitors inhibit both PARP1 and 2, and the double knockout of PARP1 and PARP2 is lethal [22].

## 3. Advanced Compound Development and Pre-clinical Studies Resulting from the Newcastle–Agouron Collaboration

Our next piece of good luck (serendipity #3) was to meet Zdenek Hostomsky at the 9^th^ NCI/EORTC annual meeting in Amsterdam in 1996. He was proposing to initiate a PARP programme at Agouron Pharmaceuticals (in La Jolla, California, USA), a company specialising in crystal structure-based drug design. He had obtained the plasmid containing the construct for the PARP1 catalytic domain from Gilbert deMurcia and the co-ordinates of its crystal structure from Georg Schultz [23]. All he needed was highly potent inhibitors. We struck up an agreement and gave him NU1085 to co-crystalise with the PARP1 catalytic domain. Initial promising results meant that their 6 month pilot funding to Newcastle was extended for a further 2 years to support two post-doctoral chemists and a technician, Alex White, Sarah Mellor and Richard Davies, and two post-doctoral biologists and a technician, Carol Delaney, Chris Calabrese and Lan Zhen Wang. At Agouron, analysis of the interactions made between the inhibitor and the PARP catalytic domain by Bob Almassy led the lead chemist on the project, Steve Webber, to propose tricyclic compounds with the carboxamide constrained within a 7 membered ring. Synthetic medicinal chemistry at Agouron was undertaken by Don Skalitzky and Stacie Canan, working with Steve Webber. Karen Maegley ran the biochemical evaluation, with cell-based target inhibition and cytotoxicity assays being done by Jianke Li. Ted Boritzki and Bob Kumpf led the conventional and computational pharmacology, respectively at Agouron in collaboration with Chris Calabrese and Huw Thomas in Newcastle. At the end of the 2 year period, several potent tricyclic indole and benzimidazole inhibitors had been made at Agouron and they extended the funding to Newcastle for a further 3 years to cover the in vitro and in vivo pre-clinical evaluation of these inhibitors, as well as the development of pharmacodynamic biomarkers until the clinical trial started in 2003. 

Extensive investigation of the cellular activity of these tricyclic compounds in terms of their ability to enhance the cytotoxicity of the DNA methylating agent, temozolomide, and the topoisomerase I poison, topotecan was undertaken in Newcastle, principally in two colorectal cell lines, LoVo and SW620 by Lan Wang and Suzanne Kyle, with Jianke Li investigating the effects in A549 lung cancer cells. My group have always maintained the highest possible stringency regarding the cultivation of cells, confirming they are mycoplasma free by routine regular testing (2 monthly intervals) and handling each cell line separately (including reagents) to eliminate the possibility of cross-contamination. Although these studies preceded routine authentication, our stocks have been authenticated since and all human cell lines are either purchased new and placed in an authenticated bank at first or second passage or authenticated and then placed in the authenticated bank. In vivo efficacy and pharmacokinetic/pharmacodynamic (PK/PD) studies were also undertaken by Chris Calabrese working with Huw Thomas and Mike Batey [24]. The first potential clinical candidate was AG14361 (1-(4-((dimethylamino)methyl)phenyl)-8,9-dihydro-2,7,9a-triazabenzo[cd]azulen-6(7H)-one: Ki < 5 nM) [25,26,27,28,29]. AG14361 caused profound sensitisation of temozolomide and was able to overcome temozolomide resistance due to defects in DNA mismatch repair [30]. The enhancement of topotecan and ionising radiation cytotoxicity in vitro was less pronounced but again the prevention of recovery from potentially lethal irradiation was substantial. In vivo radiosensitisation studies were conducted in Manchester by Kaye Williams and Ian Stratford [29]. Investigations in NIH 3T3 cells from PARP1-knockout mice enabled my PhD student, Lisa Smith, co-supervised by a topoisomerase expert, Caroline Austin, to confirm that topotecan was more cytotoxic in PARP-null cells and sensitisation by AG14361 was indeed due to PARP1 inhibition [31]. 

Investigation of in vivo activity against SW620 and Lovo xenografts revealed some interesting data. AG14361 caused an approximately 3-fold increase in temozolomide-induced anticancer activity against Lovo xenografts, but the combination caused sustained complete tumour regressions of SW620 xenografts. This was curious because in cell cultures, Lovo cells were the most sensitised to temozolomide by AG14361 and SW620 cells were not sensitised at all. SW620 cells were particularly sensitive to temozolomide alone (due to functional mismatch repair and lack of methylguanine methyltransferase). This led to another example of the synergy of science of serendipity: the discovery of the vasoactivity of this class of compounds. I reasoned that the discrepancy between the in vivo and in vitro effects meant that the tumour regression had to be something to do with the tumour micro-environment. Nicotinamide was known to be a vasodilator [32], and so I hypothesised that AG14361, which, like all PARPi contains a nicotinamide pharmacophore, might improve the delivery of temozolomide to more of the tumour cells. In collaboration with Kaye Williams and Ian Stratford in Manchester, we showed that indeed AG14361 did increase the areas perfused in the tumour, a further example of the synergy of science and serendipity (#4). [29]. This effect may have also contributed to the enhancement of the anticancer activity of the topoisomerase I poison, irinotecan, by AG14361, which was greater in vivo than in vitro. Similar vasoactive effects were reported by others with olaparib, which was accompanied by increased radiosensitivity of tumour xenografts due to better oxygenation [33]. Vasoactivity may contribute to the in vivo radiosensitisation by AG14361 but we have not investigated whether this is the case. It is possible therefore that it is a class effect and that PARPi may increase the anticancer activity of other drugs by increasing delivery to the tumour. It should be noted that these experiments were done with subcutaneous xenografts and the effect on in situ tumours may be somewhat different, especially as PARPi in the clinic have not been reported to cause hypotension except rarely in the elderly population [34], as might be expected from a vasodilator. Nevertheless, there may be effects that are specific to the tumour vasculature, but this has never been investigated.

The exciting data with AG14361 led to a proposal for a clinical trial with this compound being designed at a meeting of the team early 2000 and drafted for submission to CRUK’s New Agent Committee. The teams in Newcastle (CR Calabrese, AH Calvert, NJ Curtin, BW Durkacz, BT Golding, RJ Griffin and DR Newell), Manchester (J Monaghan, I Stratford, and K Williams) and Agouron (R Almassy, T Boritzki, S Canan-Koch, L DiMolfetto, G Furman, Z Hostomsky, A Johnston, R Kumpf, J Li, K Maegley, D. Skalitzky, SE Webber and K Zhang) all contributed to the proposal. Meanwhile, there had been continued evaluation of a number of potential inhibitors. Many of the studies that had originally been conducted with AG14361 were also replicated with several other compounds, in vitro by Suzanne Kyle and Lan-Zhen Wang and in vivo by Chris Calabrese and Huw Thomas. Computational studies by Bob Kumpf suggested a fluoro substituent at the 8 position would improve activity with the ultimate compound being 8-fluoro-5-(4-((methylamino)methyl)phenyl)-2,3,4,6-tetrahydro-1*H*-azepino[5,4,3-*cd*]indol-1-one, or AG014447. AG014447 and/or the phosphate salt—called AG014699—had been included in the pre-clinical in vitro and in vivo studies and was selected for clinical investigation [35]. AG014699 (rucaparib) was indeed more potent than AG14361 in in vivo studies and, whereas 10 mg/kg AG14361 caused complete tumour regressions in combination with temozolomide, 1 mg/kg AG014699 was equally effective [29,35]. It also had a similar vasoactive effect [36]. Pharmacokinetic (PK) studies with AG14699, conducted by Huw Thomas and Mike Batey, showed that although it was cleared quite quickly from the bloodstream, it accumulated in tumour xenografts and, at the efficacious dose, it suppressed PARP activity in the tumour by >50% for 24 h. The clinical trial proposal was re-written with AG014699 data, with input from the team above. The involvement of the team who had developed and studied the drug pre-clinically ensured that the first clinical trial was as rationally designed, with a full understanding of the pre-clinical data as it is possible to have and supported by translational pharmacodynamic, as well as PK measurements. The proposal was submitted to CRUK’s New Agent committee in 2001, with a successful outcome. AG014699, now called rucaparib, was the first PARPi to be given to a patient with cancer in May 2003.

## 4. The First Clinical Trial and Supporting Science

Based on the pre-clinical data, the target dose clinically (PARP inhibitory dose: PID) was one that inhibited PARP activity by >50% in a surrogate tissue (lymphocytes) for at least 24 h. To support this pharmacodynamically guided clinical trial, we started adapting the ^32^P NAD^+^ incorporation assay we had used to measure PARP activity and inhibition in permeabilised cells as part of the early inhibitor development. In these endeavours, we were blessed with two more items of good fortune. Firstly, experiments conducted early in 2002 by Suzanne Kyle revealed that PARP inhibition was not lost as soon as the drug was removed Figure 2A), meaning that pre-treated cells could be harvested and processed without loss of inhibition. This was an instance of real serendipity (# 5), as it is quite rare to see such persistence. Secondly, at the same time, Suzanne showed that PARP activity was maintained in cells that had been cryopreserved at -80 °C for up to 14 weeks. More importantly, rucaparib’s inhibition of PARP activity was incredibly stable: cells that had been pre-treated with rucaparib could be harvested and cryopreserved for up to 14 weeks without loss of inhibition, an even more remarkable example of serendipity (#6); Figure 2B. 

We then started harvesting lymphocytes from healthy volunteers (i.e., our colleagues in the lab). However, we soon discovered that PARP activity in human lymphocytes is orders of magnitude lower than in cultured cells and we would have needed approximately 50 ml blood pre- and post-treatment with rucaparib to be able to detect inhibition reliably. This is where yet another example of good fortune came in (serendipity #7). Alex Burkle had recently moved to Newcastle with his post-doc Ragen Pfeiffer and they had developed a much more sensitive immunoblot assay using the 10H antibody to the product, PAR [37]. They shared this technology with us and Ruth Plummer, who was my clinical fellow at that time, validated this pharmacodynamic (PD) assay to GCLP standard. It was initially intended that she would carry out the rucaparib Phase I trial. However, there was a substantial delay in the start of the trial, partially due to the take-over of Agouron by Warner Lambert (in January 1999) that was followed less than a year later by the take-over of Warner Lambert by Pfizer, who did not at that time have an oncology programme. Instead of conducting the rucaparib trial, Ruth used the PD assay to monitor the effects of temozolomide on PARP activity in patients with metastatic melanoma instead [38]. It was my next clinical fellow, Chris Jones, who conducted the trial under the guidance of Ruth Plummer and Hilary Calvert, the principal and chief investigators, respectively. Chris made further improvements to the assay and monitored PARP inhibition in the lymphocytes and tumour biopsies of patients treated with rucaparib [39]. Using this assay, another student of mine, Thomas Zaremba, set out to study PARP1 genomics, expression and activity in healthy volunteers and cancer patients with the aim of determining a) whether PARP genotype contributed to reduced PARP activity and the likelihood of developing cancer and b) whether PARP1 genotype, expression or activity contributed to unexpected toxicities in patients treated with temozolomide or radiotherapy. He did not manage to answer these questions but what he did find was that a) only 20% of the variation in PARP activity is due to the variation in PARP1 expression, b) that men have on average 40% more PARP activity than women and c) that it is androgen-driven, suggesting post-translational modification or co-activators/repressors that are androgen regulated [40]. The question this raises is: when used as a chemo- or radiosensitiser, should PARPi doses be lower in women?

The first trial of rucaparib was a dose escalation trial in cohorts of 3 patients with a variety of solid tumours receiving 1, 2, 4, 8 and 12 mg/m^2^ in combination with half-dose temozolomide (100 mg/m^2^). There was a dose-dependent decrease in PARP activity, with the desired ≥50% inhibition of PARP activity for 24 h being consistently achieved above 8 mg/m^2^. In the absence of increased toxicity, 12 mg/m^2^ was established as the safe dose to begin escalation of the temozolomide up to full dose (200 mg/m^2^) in melanoma patients consenting to a pre-treatment and post-treatment biopsy. There was a dose-dependent decrease in PARP activity in the tumours too, with approx. 90% inhibition 4–8 h after administration of 12 mg/m^2^. Escalating the dose to 18 mg/m^2^ led to increased haematological toxicity and 12 mg/m^2^ was established as the recommended Phase 2 dose to take forward in combination with temozolomide. Unfortunately, although this combination may have shown an improvement on historical data of response to temozolomide alone in patients with melanoma, there was also more haematological toxicity [41]. This may have doomed rucaparib to the same failure to progress beyond phase II studies, as with other DNA repair inhibitors such as the MGMT inhibitors O6 benzylguanine and Patrin [42], had it not been for the next and most significant example of serendipity on this project.

## 5. Discovery of the Synthetic Lethality of PARPi 

The synergy between science and serendipity (#8) reached its peak with the discovery of synthetic lethality in BRCA mutant and other cancers defective in homologous recombination DNA repair (HRR). This came about through lucky coincidences with timing of various discoveries and meetings as described below. At the end of 2001, Herbie Newell was invited to speak at Sheffield University about the drug development programme in Newcastle University. Present at this meeting was Thomas Helleday, who had been recruited to Sheffield from Tomas Lindahl’s lab. Tomas Lindahl (co-recipient of the Nobel prize for chemistry 2015 for his work on DNA repair) had a long-standing interest in PARP, its function and the effects of inhibitors. Tomas Lindahl had noted that in the presence of PARP inhibitors, there was an increase in recombination events. This led him to propose that the negative charge on the ADP-ribose polymer helped to repel negatively charged DNA from sites of DNA breakage to avoid unwanted recombination events [43]. Thomas Helleday had investigated further and found that cells lacking XRCC1 (the scaffold protein recruited to DNA breaks by ADP-ribose polymers [44], as well as those lacking PARP1, accumulated more γH2AX foci (a marker of DNA damage) and RAD51 foci (a marker of HRR). As a result of Herbie’s talk in Sheffield Thomas requested some of our more potent and selective PARPi. We gave him some NU1025 and invited him to give a talk in Newcastle to present his work in January 2002. He convincingly showed that XRCC1 defects had similar effects in increasing HRR compared to PARP1 deletion or inhibition with NU1025. This suggested the polymers were not acting to inhibit recombination but instead a failure of BER led to an increase in HRR. He proposed that cell lacking BER/PARP function would be dependent on HRR for survival and as such cells lacking HRR function may be dependent on PARP activity for survival. He tested his hypothesis in cells defective in HRR (IRS1 lacking XRCC2 and irs1SF lacking XRCC3), which were killed by concentrations of NU1025 that did not kill wt or XRCC2/3 corrected cells. 

By this time, I had widened my interest in DNA repair and as well as leading the pre-clinical biological studies on PARPi, I also led the DNA-PK and ATM inhibitor projects at Newcastle. As a result of this, I attended a meeting early in 2002, where Ashok Venkitaraman described the data supporting the key role of BRCA2 in HRR. BRCA 1 and BRCA 2 were discovered in the early 1990s as breast cancer susceptibility genes but at first it was not clear why. Their function in DNA repair began to emerge towards the end of that decade and it was confirmed that they played key roles in HRR around the turn of the millennium [45,46,47,48]. More reading around the topic convinced me that our PARP inhibitors might have therapeutic potential in BRCA mutant hereditary breast cancer, which I put to the drug development team in May 2002. The BRCA2 mutant Capan-1 pancreatic cells along with the BRCA wt BxPC-3 pancreatic cells, as control, were duly purchased and their sensitivity to AG14361 was determined by Suzanne Kyle. Frustratingly, these initial experiments, conducted in July 2002, did not show that the BRCA mutant cells were more sensitive. However, experiments that had been conducted 3 to 4 months previously in wt (AA8) and the HRD derivative (XRCC3 mutant: irs1SF) Chinese hamster ovary cells had indeed shown a very marked difference in the sensitivity of the wt and HRD cells (Figure 3). To this day, I am not sure why the data were so disappointing with the Capan-1 cells as we got more impressive results both in vitro and in vivo with this cell line using rucaparib (see later). Maybe it was the design of the experiment and the exposure period (24 h) was insufficient, as less than a complete cell cycle, or the moribund cells were lost during harvesting or maybe the BxPC-3 cells were also HRD. 

We communicated our findings to Thomas Helleday, with whom we had established a collaboration, in the autumn of 2002. By Spring 2003, he had obtained the VC8 BRCA2 mutant derivatives of Chinese hamster lung fibroblast V79 cells, which he later shared with us. Suzanne got her first result with these cells in June 2003 and they were so impressive we could hardly believe it. She replicated the result several times and confirmed that 98.5% of BRCA2 mutant cells were killed by 10 µM AG14361, a concentration that had no significant effect on the viability of either parental wt cells or the BRCA2-corrected cells. Conducting in vivo experiments with these cells proved more of a challenge. The V-C8 cells would not grow subcutaneously and, conversely, the V79 cells generated very haemorrhagic tumours that rapidly made the mice unwell such that they had to be humanely killed. However, Huw Thomas eventually succeeded in growing the V-C8 cells and the BRCA2 corrected V-C8.B2 cells intramuscularly. These experiments showed that treatment of mice with AG14361 caused a complete regression of the BRCA2 mutant tumour but growth of the BRCA2 corrected tumour was unaffected. The data generated by Suzanne and Huw went into the paper that was published in Nature in 2005 [49]. These key experiments, which have led to the current use of PARPi clinically, were actually undertaken outside of the remit of the drug development programme. This was because at Herbie Newell’s suggestion, the rest of the drug development management team at Newcastle had decided that PARP was no longer to be considered a drug development project to be discussed at subsequent meetings in June 2003 as the clinical trial (albeit in combination with temozolomide) was underway. 

The data with the BRCA mutant and HRD models, where a tumour-specific defect in DNA repair is exploited by inhibiting a complementary DNA repair pathway were indeed ground-breaking. The findings completely changed the way we think about treating cancer, from trying to overcome some property of the tumour that gave it an advantage to exploiting a vulnerability. Indeed, the search for additional examples of synthetic lethality has become somewhat of a holy grail. However, I believe PARP inhibition in HRD is unlikely to be bettered as an example of synthetic lethality for 2 reasons: (1) there is a very high level of DNA SSB, that depend on PARP for their repair, due to the endogenous ROS produced by normal metabolism, and this is increased in cancer by virtue of the associated inflammation, and (2) HRR defects are relatively common in cancer, not only due to BRCA mutations. This is yet another example of the synergy of science and serendipity (#9)

Understanding the importance of the synthetic lethality of PARPi in HRD cancers, Thomas Helleday was keen to patent our findings before publication. Following meetings with Cancer Research Technology, we undertook work to protect rucaparib and these data were included in the patent (WO 2005/012305 A2) filed 16th April 2004.

## 6. The Clinical Development of Rubraca® as A Single Agent and Predictive Biomarker Development in Ovarian Cancer 

During this time, the Phase I clinical trial of rucaparib with temozolomide was ongoing and there was clinical interaction with Pfizer who later conduct a single agent dose escalation study with rucaparib. This study was led by Hilary Calvert and Ruth Plummer, who recruited Yvette Drew, a clinical fellow, in 2006 to undertake the trial. In parallel, Yvette conducted pre-clinical and translational studies with me. The translational studies included taking blood samples from the patients before and at intervals after rucaparib adminiatration. These samples were used to determine the rucaparib PK and also PARP1 genotype, protein levels and inhibition by rucaparib in the patients’ PBMCs (lymphocytes).She also undertook studies in a panel of BRCA wt and mutant human cancer cell lines to confirm the differential sensitivity we had seen in the matched Chinese hamster cells (AA8/irs1SF and V-C8/V-C8.B2). This panel included the UACC3199, which had epigenetically silenced BRCA1 due to promoter methylation and the Capan-1 cells that had been disappointing in the AG14361 experiments. Such studies were needed because there were reports that other BRCA mutant human cancer cells were insensitive to PARPi, including AG14361 [50,51]. Fortunately, rucaparib sensitivity was found to be substantially greater in the BRCA mutant cells (including the Capan-1 cells) than the wt and, importantly, the BRCA silenced cells were comparable with the mutant ones in this respect [52]. 

At the same time, I had established an interaction with a gynaecological surgeon, Richard Edmondson, who had a number of trainees keen to undertake lab-based projects for MD or PhD. This turned out to be yet another example of the synergy between science and serendipity (#10). I was aware that BRCA mutation/HRD was a determinant of sensitivity to cisplatin/carboplatin from discussions I had had with Paul Harkin at a meeting in 2003. The standard of care for ovarian cancer is surgery followed by carboplatin + paclitaxel chemotherapy with an initial response rate of approximately 60%. Knowing that only approximately 15%–25% of ovarian cancer is associated with BRCA mutations, that left 35%–45% of the responses unaccounted for. I reasoned that they must be HRD for some other reason and that we could test this hypothesis by measuring γH2AX and RAD51 foci. γH2AX foci are formed at collapsed replication forks and DNA DSBs and RAD51 coats the SS DNA to form the nucleoprotein filament needed to invade the complementary DNA on the sister chromatid during HRR resulting in RAD51 foci. Therefore, γH2AX foci indicate the lesions generated by PARP inhibition and RAD51 foci indicate when HRR occurs. We first met with Pfizer representatives (including Zdenek Hostomsky and Gerrit Los) in September 2006 to suggest that Yvette Drew, the clinical fellow undertaking the trial of single agent rucaparib, would not only work with cell lines to establish the spectrum of rucaparib sensitivity (described above) but also to develop an assay for HRR function, based on measuring γH2AX and RAD51 foci. We also proposed to evaluate the HRR status in primary cultures of ovarian cancer ascites collected during primary/debulking surgery with another clinical fellow Asima Mukhopadhyay, a trainee gynaecological oncology surgeon supervised by Richard Edmondson and me. Thanks to Zdenek Hostomsky, ever our PARP champion even though he had been diverted to other projects, I secured funding from Pfizer in 2008 to support these studies with an additional technician, Evan Mulligan, to help with planned in vivo efficacy studies.. 

These studies led to two key papers published in 2010/11 [52,53]. The data Yvette generated in the cell lines not only demonstrated the difference in rucaparib cytotoxicity between HRD and HRR functional cells but also that RAD51 foci were a useful discriminant/potential biomarker of HRR function. That is, the induction of RAD51 foci in the HRD cells was not significant whereas there was a significant increase in the BRCA wt cells. Looking at γH2AX foci really brought it home to us how much endogenous damage cells sustain without the addition of exogenous genotoxic agents. The level of the foci after only 24 h of inhibition of endogenous damage repair with rucaparib was the same as the standard radiotherapy fraction of 2 Gy. After some significant work to establish the optimum conditions to grow primary cultures from ascites, Asima successfully translated this HRR function assay to the ascites cultures and her initial studies showed that approximately 60% of the primary cultures were indeed HRD and that this corresponded to a greater rucaparib-induced inhibition of cell growth. This was the first demonstration that >50% of ovarian cancers are HRD and, at significantly less expense than similar findings by the TCGA, which was published the following year [54]. 

The in vivo experiments in mice bearing BRCA1 mutant (MDA-MB-436) or BRCA2 mutant (Capan-1) xenografts, conducted by Yvette Drew and Evan Mulligan, showed that rucaparib caused significant tumour growth delay and this was more pronounced when the treatment period was 5x weekly for 6 weeks compared with daily x10. This was not entirely unexpected from an understanding of the underlying mechanism whereby to kill the cells PARP must be inhibited as the cell progresses through S-phase. These findings were recapitulated in the clinical study that Yvette was conducting where responses were seen when the patients switched from daily treatment for 5 days every 3 weeks to continuous dosing [55]. This study, initiated in 2007, was the first single-agent trial of rucaparib, sponsored by CRUK and Pfizer. It began very cautiously, starting at low dose (escalating from 4 to 18 mg/m2 i.v) on an intermittent schedule (daily for 5 days every 3 weeks) based on the trial in combination with temozolomide. During the study, there was a switch to an oral formulation and daily dosing, starting at 92 mg (equivalent to 18 mg/m2 based on an oral bioavailability of approximately 30% and average body surface area of 1.7 m2) once a day for 7 days and escalating to 600 mg 2x/day continuously. Many more responses were seen once dosing was continuous. It is not altogether clear whether it was the increase in the dose or the intensity of the schedule that was critical. Unfortunately, this rather slow start allowed rucaparib to be overtaken by olaparib, which went into its first clinical trial as a single agent and the dose was escalated more quickly, as there was no history of combination toxicity data to suggest a cautious approach was necessary. As a result, the first publication showing that the pre-clinical data on the synthetic lethality of PARPi in HRD/BRCA mutant cancer was translatable to the clinic was with olaparib [56]. 

In a way, this result could have been avoided with more careful scrutiny of the pre-clinical data. Pre-clinically much lower (at least 10x) concentrations (in cell cultures) and doses (in tumour xenograft studies in mice) were needed to sensitise cells and tumours to temozolomide than were needed for single agent activity in BRCA mutated/HRD cancer cells. Furthermore, increasing the dose of rucaparib to one that was active against BRCA mutant xenografts was so toxic that the mice had to be humanely killed a few days into the drug administration [29,35]. These differences are easily understood by reference to the mode of action of the PARPi in the different scenarios. Because DNA integrity is so important, the cell has more than adequate levels of repair enzymes to cope with endogenous and most environmental DNA damage. Couple this with the fact that single base lesions and SSB are the commonest form of endogenous lesions, and these are dependent on PARP for their repair, and it is easy to see that cells will have a surplus of PARP to cope with daily levels of DNA damage. In order for the damage to overwhelm this repair capacity and cause death in cells lacking the back-up pathway of HRR, there has to be a total suppression of PARP activity and relatively high levels of drug are needed to achieve this. In addition, PARP must be inhibited long enough for all tumour cells to enter S-phase with unrepaired DNA lesions and so long/continuous exposures are necessary. In the alternative scenario where, e.g., temozolomide, induces orders of magnitude greater numbers of base lesions and SSBs, it is not necessary to inhibit their repair completely to render the remaining DNA damage sufficient to kill cells and so much lower levels and shorter duration of PARP inhibition are needed. Also, when used as a single agent, the PARPi is exploiting a tumour-specific defect, and side effects are not expected so higher doses can be used. When the PARPi is synergistic with the cytotoxic in cancer cells the combination is likely to also be synergistically toxic to normal tissues so only lower doses are tolerated.

The observation that continuous therapy was much better than intermittent led us to investigate how durable PARP inhibition by rucaparib was. This study, initiated in 2011 in collaboration with and support from Zdenek Hostomsky and Gerrit Los, was partially based on our observations in the first clinical trial: as the dose of rucaparib was increased PARP activity was suppressed in PBMCs for 24 h after the 30 min infusion, even though the drug was no longer detectable in the plasma. Indeed, at the 12 mg/m2 dose, PARP activity was still way below pre-dose levels on the Monday after the last Friday dose in many patients [39]. Pre-clinically we found that rucaparib accumulated above the extracellular concentration and was retained after drug removal in cancer cells and that PARP was inhibited for at least 72 h after a 30 min exposure. In in vivo studies, a single oral dose of rucaparib inhibited PARP activity in xenografts for at least 7 days even though it was cleared from the blood within 48 h and was undetectable in the tumour within 72 h. Moreover, tumour growth inhibition following weekly dosing with rucaparib was equivalent to daily dosing [57]. This, I believe, is another example of serendipity (#11) ensuring the continuous suppression of PARP activity necessary for antitumour activity as a single agent. It is possible that this might be exploited clinically by weekly dosing schedules.

Pfizer initiated a Phase I study in combination with carboplatin alone or with paclitaxel or pemetrexed or a combination of cyclophosphamide and epirubicin NCT01009190 in November 2009 [58]. There was little pre-clinical rationale for such a study, as PARPi do not sensitise cells to either pemetrexed (an antifolate), cyclophosphamide (DNA cross-linking agent), epirubicin (DNA intercalating topoisomerase II poison) or paclitaxel (an antitubulin agent, not a DNA damaging agent) and the data with cisplatin or carboplatin are variable and cell line dependent, most probably additive in HRD cells only [59]. The triple combination arms were dropped during the study and the MTD in combination with carboplatin (AUC 5) was 240 mg one daily (carboplatin day 1, rucaparib days 1–14 of a 21 day cycle) due largely to haematological toxicities [58]. Thus, the MTD for rucaparib in combination with carboplatin (with which is does not synergise) is approximately four times higher than with temozolomide and 2.5x lower than as a single agent.

## 7. Studies That Led to FDA Approval

In June 2011, Pfizer licenced rucaparib to Clovis oncology and in November, study 10, (NCT01482715) a phase I–II trial of oral rucaparib as a single agent, was initiated [60,61]. By this time, Hilary Calvert had moved from Newcastle to UCL and he supported another clinician, Rebecca Kristeleit, who led the study. The Phase I component was a dose escalation of the oral formulation, based on the intermittent vs. continuous dosing study described above, in patients with solid tumours to establish the recommended phase II dose (RP2D) based on tolerability and efficacy. The RP2D of 600 mg 2x daily was then used to treat women with BRCA1/2 mutated high-grade serous ovarian cancer as the Phase II part of the study. The objective response rate was 60% and the toxicities were generally mild with only 5 out of 98 patients having to discontinue treatment due to an adverse event. 

Ongoing studies with other PARPi clearly demonstrated that BRCA mutation was not synonymous with a response in ovarian cancer [62], with cases with BRCA mutations in the non-responder group as well as cases without BRCA mutations in the responder group. Indeed, our earlier studies (described above) clearly indicated that BRCA mutation significantly underestimated the HRD and PARPi sensitive population. Several companies were developing predictive biomarkers of HRD and we shared some of our samples that we had previously characterised by the functional assay (by this time, our collection had grown substantially) with Tom Harding at Clovis and Foundation Medicine from 2011 to 2013. However, it seemed that the normal cellular component (that died off during culture leaving us with just the cancer cells to classify) diluted the cancer component too much for the samples to be of much use. Nevertheless, based on the observation that BRCA mutations result in substantial genome-wide loss of heterozygosity (LOH), Foundation Medicine [63] developed a genomic profiling test to identify HRD independent of BRCA mutation to be used as a companion diagnostic in the Ariel 2 trial NCT01891344 [64]. Similar tests have been developed by Myriad. Although these tests represent genetic “scarring” caused by an HRR defect at some time during the cancer’s development, but not necessarily at the time of treatment, they are substantially less challenging than functional assays, which by their nature require viable fresh tumour tissue. Screening for mutations in known HRR genes has also been used.

Ariel 2 was an innovative and pivotal trial that led to the rucaparib being given breakthrough status with the FDA. In this trial, 198 patients with high-grade serous ovarian cancer (HGSOC) were characterised: 40 had BRCA mutations (germline or somatic), 80 had high levels of LOH (presumed HRD) and the remaining 70 had neither BRCA mutation nor LOH. The median progression-free survival for the BRCA mutant and high LOH groups was significantly longer than the biomarker negative group. Accelerated approval was given by the FDA in December 2016. The follow-on Ariel 3 study led to the FDA approval of Rubraca (in April 2018) as maintenance therapy in platinum-sensitive ovarian, fallopian tube or primary peritoneal cancer, along with Foundation Medicine’s complementary diagnostic CDx _BRCA LOH_ in 2016. Ariel 3 (NCT01968213) was a randomised double-blind placebo-controlled trial in 564 patients that had responded to platinum-based chemotherapy. This trial showed a significant improvement in median progression-free survival in all patients treated with rucaparib (10.8 months) compared to placebo (5.4 months), but particularly the 130 BRCA mutated (16.6 months) and 106 BRCA wild-type, LOH-high HRD patient (9.7 months months) subgroups compared to the BRCA wt LOH low (6.7 months) [65]. More recently, Rubraca has been approved by the European Medicines Agency in January 2019 and by the National Institute for Health and Care Excellence in October 2019 for maintenance therapy in epithelial ovarian, fallopian tube, or primary peritoneal cancer who are in a complete or partial response to platinum-based chemotherapy, without the need for a companion diagnostic. Foundation Medicine’s CDx _BRCA LOH_ may still be used as a complementary diagnostic. This is because the Ariel 3 trial showed that platinum sensitivity was a sufficiently good indicator of response to rucaparib, which was indeed the basis of our hypothesis that led us to investigate HRD status in ovarian cancer in 2008.

Rubraca is in several advanced clinical trials as a single agent in other cancers associated with BRCA mutations: breast, pancreatic and prostate cancer. It has breakthrough designation with the FDA for castrate-resistant prostate cancer. It seems as though the best responses are observed in the ovarian setting. Whether this is because BRCA mutations, or even LOH are not entirely synonymous with HRD [66], or whether something within the tumour microenvironment influences the outcome is anybody’s guess at the current time. If it is the former, then our small studies on the frequency of cancers lacking HRR function, as measured in viable tumour tissue, may be relevant [67] and suggest more complex readouts of HRR function may be necessary. Combinations with other molecularly targeted agents are also being investigated in other tumour types. For most up to date data, visit https://www.clovisoncology.com/pipeline/rucaparib/ and https://clinicaltrials.gov/.

## 8. Conclusions

In conclusion, the story of the development of rucaparib/Rubraca shows not only the necessary input of intellectual (creative, deductive and analytical) and physical effort needed over a protracted period of time for the successful development of a drug (summarised in the timeline, Figure 4) but also how luck played a significant role all the way through from the initial hit right through to the clinical development in ovarian cancer. The role of luck is by no means unique to the development of Rubraca®, luck has probably contributed to many of the scientific advances that we take for granted. One prime example was the production of penicillin during the Second World War, which would not have been possible without the necessary contacts and proximity of the potteries, a fact acknowledged by Norman Heatley himself in his lecture “Penicillin and Luck” [68]. There have been some setbacks in the development of Rubraca® but, on the whole, luck has been on our side—most significantly, the fact that PARPi block the repair of the most common forms of endogenously generated DNA damage and that this is synthetically lethal with a relatively common DNA repair defect in cancer, HRR. This has caused a paradigm shift in how we think about treating cancer and there are numerous investigations to identify other instances of synthetic lethality. It seems unlikely that anything as profound as the synthetic lethality of PARPi against HRD cancers will be identified. Some examples of serendipity are yet to be exploited to the full—in particular, the discovery of the durability of PARP inhibition by rucaparib warrants further exploration clinically in pharmacodynamically guided intermittent schedules. The outcome of such a study may prove to be another paradigm shift in our handling of molecularly targeted drugs, where the biologically effective dose rather than the tolerable dose is adopted as the appropriate measure to decide the recommended phase II dose.

## 9. Patents

### 9.1. Expired Patents

Griffin, R. J., Calvert, A. H., Curtin, N. J., Newell, D. R., and Golding, B. T. Benzamide Analogues ADPRT (PARP) Inhibitors. Patent Application Number PCT/GB95/00513 (1995)

Griffin, R. J., Calvert, A. H., Curtin, N. J., Newell, D. R. and Golding, B. T. Benzimidazole PARP Inhibitors.Patent Application Number PCT/GB96/01832.

Griffin, R. J., Calvert, A. H., Curtin, N. J., Newell, D. R. and Golding, B. T. Prodrugs of PARP Inhibitors.Patent Application Number PCT/GB97/02701.5

### 9.2. Active Patents

Helleday T and Curtin NJ. Therapeutic Compounds (PARP inhibitors in homologous repair/BRCA defective cancer) Patent Application Number PCT/GB2004/003183. Publication number WO 2005/012305 A2 Divisional application 16th April 2004 GB 0408524. WO2005012305A3 

Boritzki TJ, Calvert AH, Curtin NJ, Dewji MR, Hostomsky Z, Jones C, Kaufman R, Klamerus KJ, Newell DR, Plummer ER , Reich SD, Steinfeldt HM, Stratford IJ, Thomas HR Williams KJ.. Therapeutic Combinations Comprising PARP inhibitor US application No. 60/612,458 Filed 22^nd^ September 2004 WO/2006/033006) THERAPEUTIC COMBINATIONS COMPRISING POLY(ADP-RIBOSE) POLYMERASES INHIBITOR 

## Figures and Tables

**Figure 1 cancers-12-00564-f001:**
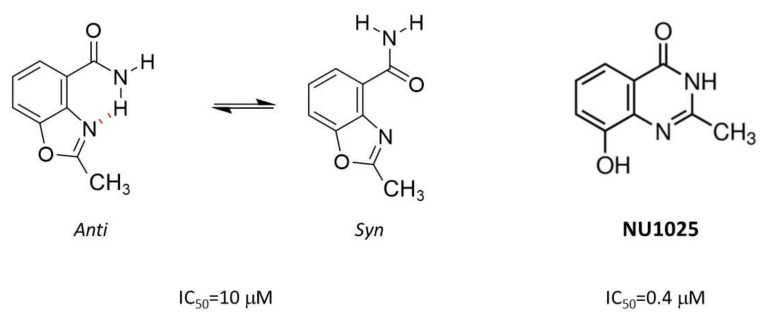
Accidental synthesis of first “hit” compound, NU1025. During the attempted synthesis of 2-methylbenzoxazole-4-carboxamide, which could exist in the desired anti- conformation or the syn- conformation (left), a molecular rearrangement occurred, resulting in NU1025, a much more potent compound.

**Figure 2 cancers-12-00564-f002:**
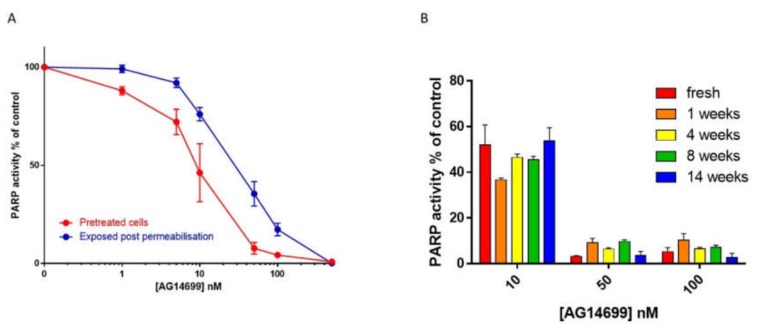
Inhibition of cellular poly (ADP-ribose) polymerase (PARP) activity. (**A**) PARP inhibition by increasing concentrations of AG14699 (rucaparib) following pre-treatment of L1210 cells (30 min) with drug prior to harvesting (red symbols and line) compared with drug added to the permeabilised cells in the reaction mixture (blue). Data are the mean and SD of three replicates from a representative experiment. (**B**) Stability of PARP inhibition by rucaparib with storage. Cells were exposed to rucaparib for 30 min prior to harvesting and cryopreservation. PARP activity was measured by ^32^P NAD^+^ incorporation. Previously unpublished data.

**Figure 3 cancers-12-00564-f003:**
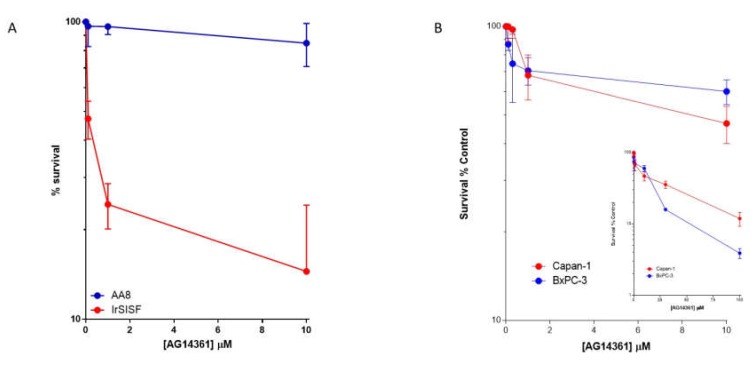
AG14361 is more cytotoxic in XRCC3 mutant irs1SF cells compared to wild-type AA8 Chinese hamster ovary cells (**A**) but BRCA2 mutant Capan-1 cells are not more sensitive than BRCA wild-type BxPC-3 pancreatic cancer cells (**B**). Cells were exposed to increasing concentrations of AG14361 for 24 h prior to seeding for colony formation in fresh medium. The insert shows an extended concentration range for pancreatic cell lines. Previously unpublished data pooled from three independent experiments.

**Figure 4 cancers-12-00564-f004:**
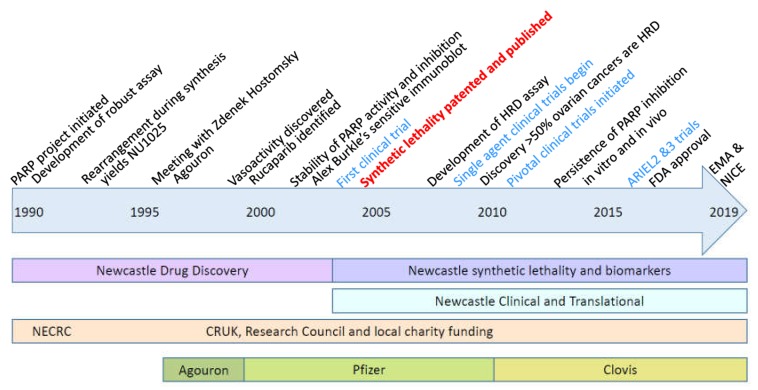
Timeline of the development of Rubraca®. Key milestones are indicated at the top of the timeline, the nature of the work undertaken at Newcastle, the funding sources and pharmaceutical companies involved are shown below the timeline.

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
