# Peer review of "The Development of Rucaparib/Rubraca®: A Story of the Synergy Between Science and Serendipity"

_cancers, 2020, doi:10.3390/cancers12030564_

Round 1
Reviewer 1 Report
This is a very interesting article, written by an insider and expert revealing very detailed insights in this incredible from bench to bedside-success story. The information are relevant and presented in a good way, so for people with interest in this drug but also beyond, such as DNA damage repair or even psychology/management regarding exemplifying the effectiveness and success of team work, the review has strong value. The importance of PARPi as a game changer in how to tackle cancer by its vulnerabilities is significant. A few comments are allowed:
How about adding a chapter in the end putting Rubraca in relationship to other PARPi. In terms of efficacy (or other pros/cons), economic success and mode of action (if there is)?
I am personally interested in how this drug reaches such durability, even after freezing. Did anyone look into epigenetic reprogramming of cells treated with Rubraca?
Regarding the synthetic lethality assays in Capan-1 cells: was there quality checks performed such as mycoplasma or STR authenticity confirmation? Unrecognized cell contamination may be the cause for perplex results, and the importance of including of rigid QM in preclinical academics labs zo increase reproducibility of results could be mentioned here.
P4 l136 PAR>PARP
Author Response
Thank you for the appreciation of my review in response to your comments:
1. Regarding a separate section comparing rucaparib with the other PARPi. I really do appreciate this suggestion but I cannot address it for 3 reasons:
- The focus is on the way both science and serendipity have contributed to the development of rucaparib and not the other inhibitors. I don’t think there is much difference in terms of mode of primary action (PARP inhibition, that is) between rucaparib, olaparib and niraparib. Talazoparib is orders of magnitude more potent and veliparib is weaker. I suspect that some of the toxicities are due to off-target effects that come with setting the approved dose at the maximum tolerated dose defined in Phase I/II studies but I don't feel confident to make such a statement in a publication
- We have never compared rucaparib directly with the other inhibitors. Certainly olaparib has overtaken rucaparib in terms of clinical evaluation but I think this is a reflection of the different priorities of AstraZeneca and Pfizer and the size of Clovis rather than anything intrinsically different between the inhibitors. Again, I don't think it would be appropriate for me to make a public statement to this effect
- The review already exceeds the standard word-limit and I’m reluctant to push my luck any further
2.
I don’t think the durability is anything to do with epigenetic reprogramming, rather I think it is to do with the KD relative to the Km for NAD+ and the drug is just very sticky (essentially irreversible without astronomic intracellular [NAD]). My hunch is that the time taken for PARP activity to recover is a reflection of the turnover and synthesis of new PARP1 protein molecules after free drug has been washed away. I have not tested this though as I think the cyclohexamide treatment would need to be so long that the cell would die long before a return of PARP activity could be seen in the controls.
3.
Actually my first 2 years as a post-doc were completely wasted due to my predecessor’s cross-contamination so I have always been obsessively careful with cell culture. I initiated mycoplasma testing, and quarantine handling of cells from outside the department (then the Cancer Research Unit) in the early 1990s, long before it was routine practice in other labs. Back then it was Hoechst 33258 staining and hours looking down a microscope but we have since used a variety of quicker methods, PCR and now ELISA. It is so long ago that we did these experiments (2002) that I can’t remember what test we were using at the time. The Capan-1 cells were bought from ATCC immediately before we did those experiments and they were authenticated (STR profiled) in 2014.
I have added the following text page 4/5 l 169-174
My group have always maintained the highest possible stringency regarding the cultivation of cells, confirming they are mycoplasma-free by routine regular testing (2 monthly intervals) and handling each cell line separately (including reagents) to eliminate the possibility of cross-contamination. Although these studies preceded routine authentication our stocks have been authenticated since and all human cell lines are either purchased new and placed in an authenticated bank at first or second passage or authenticated and then placed in the authenticated bank.
However, I’m not sure it is necessary
4
P4 l136 PAR>PARP
I actually meant PARP1 enzyme here, so I have clarified in the text
Reviewer 2 Report
Hi, the author of the review paper provided a personal account of the story behind the development of the rucaparib/Rubraca and highlighted serendipity is a contributing factor to the success story. Sufficient scientific references were accurately cited. The rationale and reasoning behind some difficult, unresolved questions were discussed as first person narrative. Given this review is in part a personal account of the story so it is within the scope of the work. The review is well-written, enjoyable to read, scientific sound. Minor comments are as follows:
- Line 23: lucky accidents? Maybe, unexpected turns and fortunate outcome
- Line 24: history of the relationship between science and serendipity…. I felt there is no relationship between science and serendipity. Maybe this review describes the history behind the research that took place and how serendipity played a major role that brought us to the current position when it comes to the clinical utility of rucaparib in cancer treatment?
- Line 37: it is hard to avoid the conclusion that luck has played a significant role…instead of luck… maybe serendipity is a better choice of wording?
- Line 43: exposure of cells to methylating agents? DNA methylation of protein methylation agents?
- Line 61: really got ‘inhibitor development’ going? Could consider ‘small molecule drug development’ going instead?
- Line 61: They establish a drug development programme…. Did the programme started with high throughput screening? Phenotypic or target based screening?
- Line 83: The authors repeated use the attribute success to serendipity but maybe there is also component of persistence involved in the success. Despite difficulties, persistence pays off.
- Line 111: She identified it was the DNA repair phase… what is the DNA repair phase? Part of a cell cycle? Or a specific time during which cells having DNA repair activated? Can this be further scientifically specified?
- Line 112: …. was critical as NU1025 was as active when it was added after a 20 min pulse with MTIC as when the 2 compounds were given simultaneously…. I felt that this sentence needs to be rewritten. The message is rather unclear.
- Line 127: The generation of PARP knockout mice that were both viable and fertile by 3 groups in the late 1990s…. The generation and characterisation of PARP knockout mice by 3 groups in the late 1990s independently showed that the mice were both viable and fertile.
- Line 166: DNA methylation agent? or Alkylating agent?
- Line 198: but we have not investigated…. Why not? How could this be achieved?
Other comments: given this work is in part a personal account. Perhaps the authors should consider including some photos of the investigator involved with the work.
Author Response
Thank you for your appreciation and for taking the time to look at my review.
In response to your minor comments:
- Line 23: lucky accidents? Maybe, unexpected turns and fortunate outcome I have changed this to fortuitous occurrences (the reason for “accidents” was because of the spontaneous rearrangement during the synthesis of NU1025)
- Line 24: history of the relationship between science and serendipity…. I felt there is no relationship between science and serendipity. Maybe this review describes the history behind the research that took place and how serendipity played a major role that brought us to the current position when it comes to the clinical utility of rucaparib in cancer treatment? Whilst I appreciate the comment I feel that the text is OK as it is in the abstract, which is word-limited. Each step in the progress of the development of the drug has been dependent on science or serendipity (sometimes both together) and that the ultimate success has been dependent on both. Science on its own would not have led to the successful outcome if there hadn’t been a relationship with these lucky coincidences. This is what I hoped to bring out in the text but it would unnecessarily lengthen the Abstract
- Line 37: it is hard to avoid the conclusion that luck has played a significant role…instead of luck… maybe serendipity is a better choice of wording? I have looked up both luck and serendipity in the OED online. Luck is “good things that happen to you by chance, not because of your own efforts or abilities“ serendipity is “the fact of something interesting or pleasant happening by chance” I therefore regard them as largely interchangeable but luck being marginally closer to what I wanted to say here.
- Line 43: exposure of cells to methylating agents? DNA methylation of protein methylation agents? This has now been changed to ethyleneimine for clarity
- Line 61: really got ‘inhibitor development’ going? Could consider ‘small molecule drug development’ going instead? I am not sure that this change is strictly necessary
- Line 61: They establish a drug development programme…. Did the programme started with high throughput screening? Phenotypic or target based screening? I believe the subsequent text (lines 64-76) explains this sufficiently. PARP was 1 of 3 targets and the subsequent text regarding the production of a reliable assay based on references 7 and 8 makes it clear that it was not a high throughput assay
- Line 83: The authors repeated use the attribute success to serendipity but maybe there is also component of persistence involved in the success. Despite difficulties, persistence pays off. I guess strictly-speaking it wasn’t a chance happening that plastic tubes were the solution but it was a chance happening that Barbara Durkacz and her technician (who was doing the assay) were both on holiday at the same time and I could set to and sort it out without interfering with them. I did not want to draw attention to their inability to solve it themselves
- Line 111: She identified it was the DNA repair phase… what is the DNA repair phase? Part of a cell cycle? Or a specific time during which cells having DNA repair activated? Can this be further scientifically specified? Now line 112 I have added the words “rather than the DNA damage induction phase” for clarification
- Line 112: …. was critical as NU1025 was as active when it was added after a 20 min pulse with MTIC as when the 2 compounds were given simultaneously…. I felt that this sentence needs to be rewritten. The message is rather unclear. See above
- Line 127: The generation of PARP knockout mice that were both viable and fertile by 3 groups in the late 1990s…. The generation and characterisation of PARP knockout mice by 3 groups in the late 1990s independently showed that the mice were both viable and fertile.The text has been changed as suggested
- Line 166: DNA methylation agent? or Alkylating agent? No methylating agent is more accurate as temozolomide methylates DNA alkylating agent might suggest it could also ethylate, chloroethylate etc. Inded some people refer to cisplatin as an alkylating agent when it does not attach alkyl groups
- Line 198: but we have not investigated…. Why not? How could this be achieved? I think it would be difficult to distinguish radiosensitisation that was due to vasoactivity as opposed to inhibition of radiation-induced DNA damage
Other comments: given this work is in part a personal account. Perhaps the authors should consider including some photos of the investigator involved with the work.
I think I might run into problems with permissions etc.
Reviewer 3 Report
The MS by Nicola J Curtin tells the story of the development of rucaparib from an insider’s perspective. The author was a key figure in the Newcastle team that succeeded, through various difficulties as detailed in the paper, in bringing this PARP inhibitor drug to the market. The storytelling is captivating and the behind-the scene-details make the text really enjoyable to read. Even as a fellow PARP enthusiast the reviewer found various pieces of information novel, interesting and worthy of notice. The MS is well structured, informative and the many personal aspects the author shares with the reader make the paper stand out from the long series of typical uniform reviews written in an alienating neutral tone.
Minor points:
In line 31: “Poly(ADP-ribose polymerase” should be “Poly(ADP-ribose) polymerase” Figure 2. presents the PARP inhibitory profile of rucaparib. On the “x” axis of panel “A” the drug’s code should be corrected from “AG140699” to “AG14699”. Moreover, if these are published data, the source should be cited. If these are unpublished data, some experimental details should be provided, “n” numbers and meaning of error bars (SD or SEM) should be specified along with statistical analysis (what is significant and how significance was determined.) Figure 3. See comments under #2. except the note on “n” numbers which is given here.

Author Response
Thank you for your very nice comments, I was worried how the different style would be received.
Regarding the minor comments I have corrected the typo in the text and figure 2. I have included the relevant data in the figure legends (all data have not been published before. WIth regards to Figure 2: We did not do any statistical analysis here as these studies were purely to show that there was no loss of inhibition when cells were harvested and assayed after they had been exposed to rucaparib. The fact that there was greater inhibition than when drug was added to the permeabilised cells is partly explained by our subsequent findings that rucaparib accumulates within cells to a higher concentration than the extracellular one (reference 57)